# Efficacy of Photobiomodulation Therapy in Older Adults: A Systematic Review

**DOI:** 10.3390/biomedicines12071409

**Published:** 2024-06-25

**Authors:** Lidvine Godaert, Moustapha Dramé

**Affiliations:** 1EpiCliV Research Unit, Faculty of Medicine, University of the West Indies, Fort-de-France 97261, Martinique; godaert-l@ch-valenciennes.fr; 2General Hospital of Valenciennes—Valenciennes Hospital, Department of Supportive Care in Oncology, 114 Avenue Desandrouin, F-59300 Valenciennes, France; 3Department of Clinical Research and Innovation—CS 90632, University Hospitals of Martinique—Pierre Zobda-Quitman Hospital, Fort-de-France 97261, Martinique

**Keywords:** photobiomodulation therapy, older adults, efficacy, complementary treatment

## Abstract

Background: The aim was to determine whether there is any available evidence on the efficacy of photobiomodulation therapy (PBMT) in older adults. Methods: A literature search was performed including all articles published up to February 2024. Studies reporting data on PBMT in older adults were included. This study was registered with PROSPERO. Results: In total, 406 studies were identified. After eliminating duplicates and irrelevant studies, 10 records were included in the final review. In all included studies, the protocols used to deliver PBMT were different in terms of type of device, wavelength, irradiation duration, and pulse frequency. In neurodegenerative diseases, two studies reported non-significant results, while two studies reported efficacy of PBMT. In wounds and ulcers, two out of three studies reported efficacy of PBMT. In macular degeneration, one study reported efficacy of PBMT. One study on hyposalivation reported efficacy of PBMT. Conclusion: PBMT appears to be a promising complementary treatment. All studies reported good compliance and safety throughout the treatment. In the future, it will be essential to harmonize PBMT parameters. Further studies are warranted to define the best indications, the most effective protocols, and the right population to target for use in routine practice.

## 1. Introduction

Photobiomodulation first emerged in the 1960s and corresponds to the set of biological, non-thermal, non-cytotoxic effects prompted by the exposure of tissues to sources of non-ionizing light energy in the visible and near-infrared spectrum [1,2]. Initially termed “low-level laser therapy” (since lasers were the first light sources available), the term photobiomodulation is now preferred to encompass the advent of light-emitting diodes (LED) as a light source and to take account of the fact that its biological effect is related to the capacity to modulate certain cellular signals [3].

Photobiomodulation consists of exposing damaged or infected tissue to one or several wavelengths for a specific period of time, to enable transfer of energy to cell receptors called chromophores, of which one of the most well-known is the cytochrome C oxidase. This transfer of energy enables activation of the chromophore, mediating a series of cellular reactions (stimulation, modulation, inhibitions) that together bring about positive biological effects (antalgic, anti-inflammatory, neuroprotective, vasodilating). These effects have been widely demonstrated in vitro and in vivo in animal models [2,4,5]. In humans, the body of evidence is constantly growing, in both healthy volunteers and patients [6,7,8].

Widely used in aesthetic medicine, the indications for photobiomodulation therapy (PBMT) are progressively widening in line with the growth in scientific research that is expanding our knowledge of the effects in humans and enhancing our understanding of how best to harness these positive effects (optimal equipment, wavelength, dose, etc.). Recently, PBMT has been recognized as a therapy recommended for the prevention and treatment of mucositis secondary to anti-cancer treatments [7]. PBMT is also being discussed as a possible approach for disease modification in Alzheimer’s disease by the EU/US CTAD Task Force [9]. In light of these developments, the role of photobiomodulation in the care of individuals aged 65 years and over is garnering increasing interest. Several studies attest to the safety of PBMT, and there are practically no contraindications, making it an attractive therapeutic option in many indications [8].

In this context, we aimed to perform a systematic review of the literature to assess whether there are available data confirming the positive effects of PBMT in individuals aged 65 years and older, in any and all indications.

## 2. Methods

The research question to be answered by this systematic review was to determine whether there is any available evidence on the efficacy of PBMT, used for any purpose, in older adults.

### 2.1. Search Strategy

A comprehensive literature search was performed in Scopus^©^ and PubMed^©^. The literature search covered all publications up to and including 1 February 2024, with no specific start date specified. The search algorithm was defined by the two senior researchers (LG, MD) and included the following keywords in the title and/or abstract: “photobiomodulation” OR “low level light therapy” OR “low level laser therapy” OR “low power laser therapy” OR “transcranial laser stimulation”. Filters were applied to select studies in the English or French language, in human beings only, and the following publication types were excluded: reviews, case reports, editorials, and correspondence. Additional studies were searched from reviewing the reference lists of retrieved studies. The authors of the studies were contacted to recover unpublished data when available. Study selection was performed following the PRISMA (Preferred Reporting Items for Systematic Reviews and Meta-Analyses) guidelines [10]. This review was registered with PROSPERO under the number CRD42024504081.

### 2.2. Study Selection Criteria

Study eligibility criteria were defined before performing the literature search according to the PICOS framework (Population, Intervention, Comparator, Outcome, and Study design). Studies were eligible for inclusion if they reported data on PBMT and if the mean age of the study population minus the lower boundary of the standard deviation of the mean age was greater than or equal to 65 years. The intervention (exposure) was PBMT, and the comparator was the absence of PBMT. The outcome was efficacy of PBMT (clinical results). All comparative study designs were considered in this systematic review. Correspondence, editorials, reviews, basic science articles, and case reports as well as case series of fewer than 10 subjects were excluded.

### 2.3. Data Extraction

Data analysis was performed using Covidence systematic review software^©^ (Veritas Health Innovation, Melbourne, Australia) available at www.covidence.org (accessed on 24 February 2024). After eliminating duplicates, the two senior researchers independently reviewed the titles and abstracts of all articles. In case of disagreement about whether or not to include an article, the case was discussed until a consensus was reached. The researchers independently extracted the data using the same data extraction form. The following data were extracted: study characteristics (publication year, country, study design, sample size, mean and/or median age, and medical condition), PBMT parameters [3] (device used, power output, irradiation duration, dose per treatment beam spot size, pulse frequency, target location, number of sessions, and interval between sessions), and outcomes and results of the studies. When appropriate, the authors were contacted for missing PBMT parameters.

### 2.4. Quality Assessment

The quality of the included studies was assessed using the Newcastle–Ottawa scale (NOS) for observational studies [11] and the Cochrane library Risk of Bias (RoB) tool [12].

## 3. Results

In total, 406 studies were identified by the literature search (Figure 1). Among these, 26 duplicates were excluded. After examining the titles and abstracts of the remaining 380 studies, 95 articles were retained for full-text assessment. After reading the full text of these 95 studies, 85 were excluded due to wrong study population or wrong study design. Thus, 10 studies were included in the final review [13,14,15,16,17,18,19,20,21,22].

Table 1 summarizes the characteristics of the studies included in the review. Six studies were randomized controlled trials, while four were observational studies. Sample sizes ranged from 8 to 86 subjects, and mean/median age ranged from 72 to 85 years. Overall, the studies focused on four groups of medical conditions: neurodegenerative diseases, wounds and ulcers, macular degeneration, and hyposalivation. The characteristics of the photobiomodulation treatment are described in Table 2, grouped by medical condition for easier perusal.

In all studies included in this systematic review, the protocols used to deliver photobiomodulation were different in terms of the type of device, wavelength, irradiation duration, pulse frequency, etc. The outcomes and results are described in Table 3, grouped by medical condition. In neurodegenerative diseases, two studies reported non-significant results [13,15], while two studies reported positive results with photobiomodulation [17,18]. In wounds and ulcers, two [14,21] out of three studies reported positive results on healing in the population with photobiomodulation treatment, and the third [22] reported no significant results in terms of absolute or relative wound size reduction. In macular degeneration, one study reported positive results [20] and one found no significant results [16]. We found only one study investigating the efficacy of photobiomodulation in hyposalivation [19], and the results were significant in favor of PBMT.

Regarding the observational studies (Table 4a), the quality was considered high in all four studies. Concerning the randomized controlled trials (Table 4b), four were at low risk of bias, and two had a high risk of bias, notably due to issues relating to the measurement of the outcome and selection of the reported result.

## 4. Discussion

This systematic review identified 10 studies assessing the efficacy of PBMT in people aged 65 years or over [13,14,15,16,17,18,19,20,21,22], of which four were observational studies [14,18,20,21], and six were randomized trials [13,15,16,17,19,22]. These studies investigated four medical conditions overall, namely neurodegenerative diseases, wounds and ulcers, macular degeneration, and hyposalivation.

We identified four studies investigating the efficacy of PBMT in neurodegenerative diseases in the aged population [13,15,17,18]. One of them [13] reported on the effect of PBMT in Parkinson’s disease, two [15,17] reported on the effect of PMBT in cognitive disorders, while the fourth [18] reported on PBMT in vascular Parkinsonism and in Binswanger’s disease. Three authors [13,15,17] tested a transcranial photobiomodulation device, while one [18] tested a transcatheter intracerebral photobiomodulation laser to dispense photobiomodulation. Herkes et al. [13] tested the safety and feasibility of a transcranial photobiomodulation device to reduce motor signs of Parkinson’s disease, as assessed with the modified Movement Disorders Society revision of the Unified Parkinson’s Disease Rating Scale part III assessment (MDS-UPDRS-III). Parkinson’s disease is one of the most common neurodegenerative diseases in the world. People suffering from Parkinson’s disease generally present motor disorders as well as non-motor signs [23]. From a pathophysiological point of view, Parkinson’s disease is characterized by the progressive death of certain neurons, particularly dopaminergic cells in the substantia nigra pars compacta of the midbrain [24]. In the study of Herkes et al. [13], at the end of the treatment, there was no significant difference between the active and sham groups. The energy dose administered during PBMT was low (5.4 and 10.4 Joules/cm^2^), which may partly explain these results. Indeed, the energy dose administered to the target tissue is the most important parameter in PBMT. As discussed by Li et al. [25], many factors affect light propagation in the human brain (absorption, reflection) and these must be taken into account for PBMT protocols with transcranial devices. Light energy is transported by the wavelength, and shorter wavelengths have lower tissue penetration. In PBMT protocols targeting neurodegenerative diseases, it is very important to use the right wavelength with the pertinent power output and duration of irradiation in order to achieve an effective energy dose in the right cerebral region. A wavelength of 808 nm seems to have the best penetration. Herkes et al. [13] used two different wavelengths (635 and 810 nm) and a transcranial device. It is possible that intracerebral target tissues did not receive an optimal level of energy. Many studies have reported the positive effect of PBMT with the near infrared (NIR) wavelength for Parkinson’s disease, particularly in animal models and in some studies in humans [24,26]. McGee et al. [27] reported a post hoc analysis of motor outcomes in the population studied by Herkes et al. [13] and showed that some people were “good responders” to PBMT, with improvements on all sub-scores of the MDS-UPDRS-III. PBMT induces neuroprotection in animal models and appears to be able to improve abnormal neuronal activity caused by Parkinson’s disease. However, the mechanism by which photobiomodulation mediates this effect remains unclear. Mitochondria are a prime target [2,4,28]. As a first step, photobiomodulation induces an intracellular action, notably via the activation of cytochrome C oxidase, which in turn triggers a cascade of favorable reactions (increased ATP production, dissociation of NO from its binding sites, regulation of oxidative stress), ultimately improving cellular function [5] by increasing the expression of GDNF and regulation of genes associated with cell signaling, amongs others [24]. Other photoreceptors absorb at other wavelengths and have other specific biological effects [2]. Secondly, it is likely that exposure to certain wavelengths leads to changes in the extracellular environment, which in turn generate neuroprotective effects [2]. PBMT increases cerebral blood flow and oxygen availability. Long-lasting positive effects have been observed after only one light exposure, due to modulation of long-term expression of various proteins [29]. Studies suggest that exposure to NIR enables healthy cells to resist better, but also enables weakened cells to repair themselves [1].

Two authors [15,17] investigated PBMT in the setting of cognitive disorders. Blivet et al. [15] studied patients with Alzheimer’s disease at mild to moderate stage in a double-blind randomized sham-controlled study. The photobiomodulation device consisted of a helmet associated with an abdominal panel. Compliance was defined as “very good”. No significant difference was observed between the two groups for the absolute change in MMSE score or ADAS-Cog. At the end of the treatment, the changeover baseline in the ADAS-Cog comprehension sub-score was better in the active group than in the sham group (*p* = 0.029). Executive function and verbal memory tests were also improved in the active group (*p* = 0.012). Chao et al. [17] studied patients with dementia or Alzheimer’s disease at mild or moderate stage in a randomized controlled study. At the end of follow-up, ADAS-Cog and NPI-FS scores improved significantly in the photobiomodulation group but not in the control population (respectively, *p* = 0.007 and *p* = 0.03). Chao et al. [17] used transcranial and intranasal devices. As demonstrated in other studies, using the transphenoidal approach with the adapted wavelength (808 nm) seems to provide the best results [29]. The results observed in the studies by Blivet et al. [15] and Chao et al. [17] are consistent with those of other studies in younger adult populations. Saltmarche et al. [30] reported a significant improvement in the MMSE score in patients with mild to moderately severe dementia or Alzheimer’s disease treated by transcranial and intranasal photobiomodulation (*p* < 0.003) (i.e., the same device as used by Chao et al. [17]). Nagy et al. [31] reported a positive effect of photobiomodulation by nasal probe combined with aerobic exercise in an adult population with Alzheimer’s disease and anemia. Chan et al. [6] showed that visual memory performance was improved after transcranial photobiomodulation (*p* = 0.05) in adults with mild cognitive impairment. As for Parkinson’s disease, the precise mode of action of transcranial photobiomodulation in cognitive disorders is not yet fully understood. Other photoacceptors exist in addition to CCO. Nanostructured water located in heat-gated ion channels may also absorb photons and modulate the cellular or nuclear response. Each photoreceptor reacts preferentially in a given range of wavelengths [2]. Pruitt et al. [32] measured changes in cerebral metabolism in younger and older healthy subjects before, during, and after transcranial photobiomodulation. They reported that transcranial photobiomodulation improved cerebral metabolism probably via increased NO production, which may lead to vasodilation [4]. In a systematic review, the authors reported that photobiomodulation can reduce the accumulation and size of amyloid beta (aβ) protein in different brain regions in animal models [33]. An anti-inflammatory effect exerted by decreasing reactive oxygen species and by inhibiting cyclo-oxygenase 2 has been proposed as another possible mechanism of action. In addition, PBMT seems capable of activating the microglia into the M2 phenotype. This phenotype is in turn capable of exerting anti-inflammatory effects, promoting the elimination of debris and enhancing tissue repair [4]. Several authors have observed that local application of photobiomodulation can lead to beneficial effects in distant tissues [1]. These “distant” beneficial effects could be due to signals transmitted secondarily by directly stimulated structures. Blivet et al. [15] associated an intracranial device with an abdominal belt to exert action on the microbiome. The microbiome can influence metabolism and contribute to the development of certain diseases, including Parkinson’s or Alzheimer’s disease, through breakdown of cellular tight junctions that may lead to bowel permeability, allowing toxins such as LPS or bacteria themselves to enter the circulation and potentially cross the blood–brain barrier [34]. According to Liebert et al. [35], PBMT can modify the microbiome and re-establish diversity. These changes could help slow the progression of neurodegenerative diseases.

Maksimovich et al. [18] studied intracerebral transcatheter laser PBMT in patients with Binswanger’s disease or vascular Parkinsonism (intracerebral catheter versus conservative treatment). They reported complete or incomplete recovery of mental and motor functions in 92.5% of patients with PBMT versus 18.4% of patients with conservative treatment. Binswanger’s disease and vascular Parkinsonism are the result of impaired cerebral vascularization with disseminated subcortical atherosclerotic lesions. Maksimovich et al. [18] reported an improvement in blood flow in the patients receiving PBMT in their study. The neuroprotective effect of photobiomodulation has been discussed above. PBMT has shown effects on the vascular system in animal models. It could induce the release of NO, which stimulates vasodilatation [36]. PBMT is also known to stimulate damaged endothelial cells [1] and improve the secretion of angiogenic proteins in mice [36].

Three studies reported results on the efficacy of PBMT in wounds and ulcers in people aged 65 years and over [14,21,22]. Degerman et al. [14] reported an observational study about patients with hard-to-heal venous leg ulcers treated by photobiomodulation associated with traditional dressing, compared to a control group (traditional dressing alone). Healing time was reduced in the intervention group compared to that in the control group (*p* = 0.0002). Saltmarche et al. [21] also reported positive results with PBMT in a population of nursing home residents with chronic and acute wounds. Push score was used to determine the efficacy of PBMT. At the end of the study, 42.9% of the wounds were closed, and 19.0% presented a significant percentage of closure. No difference was observed between acute and chronic wounds. In patients with a decubitus ulcer, Lucas et al. [22] found no significant difference between patients receiving PBMT and a control group not receiving PBMT, in terms of either absolute or relative wound size reduction.

Due to differences in treatment protocols, it remains difficult to compare these studies. Nevertheless, it is possible to compare the energy dose administered and the wavelength used. The energy dose used was 1.0 J/cm^2^ for Lucas et al. [22], 2.4 J/cm^2^ (associated with 0.6 J/cm^2^ on intact skin) for Degerman et al. [14], and varied from 1 to 6 J/cm^2^ for Saltmarche et al. [21]. The wavelength used by Lucas et al. [22] was 904 nm. Two wavelengths, namely 904 nm and 635 nm, were used by Degerman et al. [14], and 785 nm in Saltmarche et al. [21]. According to a systematic review by Petz et al. [37], a wavelength of 660 nm and a dose of 2–4 J/cm^2^ are probably the optimal parameters for chronic ulcers. Mathur et al. [38] demonstrated in diabetic patients that PBMT associated with conventional therapy improved granulation after 15 days of treatment compared with that in a control group (*p* < 0.001). PBMT is probably an attractive treatment as an adjunct to conventional therapy. It could improve blood flow in damaged tissue, has an anti-inflammatory effect, and helps to reduce pain, according to the activated photoreceptor [39]. PBMT increases fibroblast proliferation. Positive effects on pain from ulcers were probably due to activation of opsin, a chromophore suspected of being involved in pain signal modulation, for example [40]. However, human skin expresses different types of opsin in the various subpopulations of dermal cells, and the specific role of each type of opsin remains unclear.

Two studies reported on the efficacy of PBMT and dry-age related macular degeneration (dry-AMD). Markowitz et al. [16] treated patients in a double-masked randomized, sham-controlled, single-center study of PBMT. No positive effect was observed in terms of the primary outcome, but the authors reported a statistically significant improvement with PBMT among patients with earlier stages of dry-AMD (*p* < 0.005). Merry et al. [20] found similar results at three weeks and three months, with a statistically significant improvement in the PBMT group (*p* < 0.001). The existence of “good responders” to PBMT is likely, and has been observed in other studies, although data are lacking to provide explanations for this phenomenon [27]. Based on current knowledge of the mode of action of PBMT, it is likely that the less cell or tissue damage, or the more acute the damage, the greater the chances of recovery under the influence of PBMT. There were more subjects with early stage disease (AREDS 2-3 versus AREDS 4) among the subjects studied by Merry et al. (*p* < 0.001) [20]. According to the literature, the optimal level of energy in dry-AMD would be equivalent to 4.8 J/cm^2^ at 670 nm. With this in mind, and taking into account the energy dose administrated by Markowitz et al. [16], it is possible that the absence of any significant result on the primary outcome was due to an excessive energy dose (respectively 20.8, 54.16, and 6.7 J/cm^2^). This illustrates the idea of a “biphasic dose response”, whereby low doses are stimulatory, while higher doses are counterproductive [5,28].

In our systematic review, we found one article that investigated the efficacy of PBMT on hyposalivation in aged people [19]. Brzak et al. [19] studied different wavelengths (685 nm and 830 nm) to identify the most effective protocol in hyposalivation. The primary outcome was salivary flow rate. Both laser wavelength groups presented positive results (*p* = 0.0019 and *p* = 0.0044, respectively), but patients in the 830 nm laser wavelength group had a higher salivary flow rate. Expert consensus opinion from the World Association for PhotobiomoduLation Therapy (WALT) [7] confirmed the utility of PBMT in hyposalivation or xerostomia induced by radio- or chemotherapy. Several devices are available for the administration of PBMT, such as intra-oral devices (lollipop or oral pad) or extra-oral devices (for transcutaneous stimulation of the salivary glands or parotid stimulation). The dose required to obtain a preventive or curative effect, depending on the equipment, is well established (preventive dose, intra-oral device, wavelength 650 nm: 5.7 J/cm^2^; curative dose, intra-oral device, wavelength 650 nm: 11.4 J/cm^2^; curative dose, transcutaneous device, wavelength 810 nm: 9 J/cm^2^) [7]. The mechanisms behind the action of PBMT in stimulating the salivary glands are based on an improvement in cellular function via the activation of chromophore-like cytochrome C oxidase.

“Administer the right wavelength at the right dose to the right place” is a principle that cannot be ignored in PBMT. The wavelength is very important [4], but the delivery time, power output, or the number of sessions are also important to determine the energy dose. The light signal is progressively depleted as it moves away from its source, and it depletes all the more rapidly when the material or tissue through which it passes is dense, such as skull bone. Li et al. modelled a human brain to study the diffusion of photobiomodulation at different wavelengths [25]. Their work established that the wavelength was more important than the beam type. To achieve a positive effect in cognitive disorders, for example, the optimal wavelengths are probably around 810 nm and higher [4,5,6,41]. “Continuous wave” or “pulsed light” is another technical question that must be considered. In a review by Hashmi et al. [42], pulsed light seemed to be more effective than continuous wave light to cure wound healing, pain, and ischemic stroke. Although an important parameter in determining the depth of penetration is the wavelength (shorter wavelengths have lower tissue penetration), it seems that pulsed light penetrates more deeply into the tissue than continuous wave light. Pulsed light enables the use of longer wavelengths, thus limiting surface tissue heating [42], and can be useful for deeper organ treatment. Calculating the precise dose received by a target organ is the cornerstone in PBMT protocols and should be easier in the future. Currently, however, few research teams are equipped to do so. Overall, the ideal parameters for each indication are not yet known. The WALT has done a great deal of work in the field of supportive care to publish standardized care protocols [7]. Its authors have established several protocols or recommendations for preventive and curative management of chemo- or radiation-induced mucositis. The same work needs to be progressively carried out for other indications, based on further randomized controlled studies. Hamblin [3] recently published an article entitled “How to write a good photobiomodulation article”, inviting authors to harmonize the information included in articles about photobiomodulation (e.g., energy dose, irradiation time, power output, beam size). These recommendations are an important step forward in achieving harmonized implementation and presentation of research in this field.

This work has several strengths. The number of articles initially found was high, and the articles finally selected were of quite good quality. Only four medical conditions were involved, enabling several studies to be carried out in each area (with the exception of hyposalivation, where only one study was included). Despite the great diversity of the protocols, most of the important PBMT parameters were identified [3]. Conversely, this work also has some limitations. The studies are characterized by small sample sizes, a diversity of protocols (wavelength, power output, irradiation duration, pulse frequency), and different primary outcomes despite similar indications. Comparisons between studies, even in similar indications, were therefore difficult. These discrepancies also preclude a meta-analysis of the efficacy of PBMT. The duration of patient follow-up was generally short and did not provide data on the medium- and long-term effects of PBMT in the indications studied. There were very few randomized studies with a high level of evidence.

In summary, PBMT appears to be a promising complementary treatment. All studies reported good compliance and safety throughout the treatment [8]. In the future, it will be essential to harmonize PBMT parameters. Further studies are warranted to define the best indications, the most effective protocols, and the right population to target. Future studies should adopt a randomized controlled design to ensure a high quality of evidence that could form a basis for standardized treatment protocols for use in routine practice.

## Figures and Tables

**Figure 1 biomedicines-12-01409-f001:**
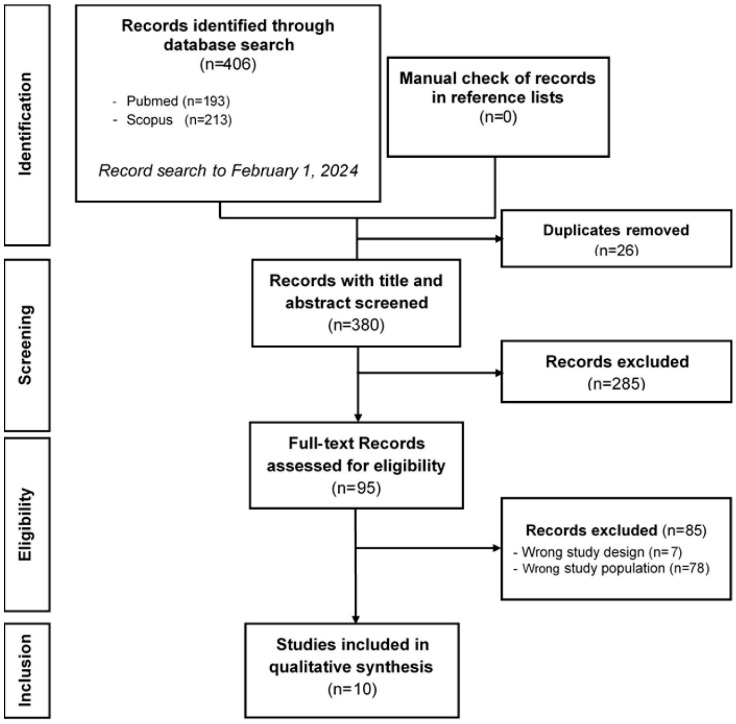
PRISMA flow diagram of the records included in the systematic review.

**Table 1 biomedicines-12-01409-t001:** Description of the 10 studies included in the systematic review.

Author, Year Ref	Country	Study Design	Sample Size	Age	Medical Condition
Herkes, 2023 [13]	Australia	Controlled trial	40	72 ± 5 *	Parkinson’s disease
Degerman, 2022 [14]	Sweden	Cohort	68	81 ± 7 *	Venous leg ulcer
Blivet, 2022 [15]	France	Controlled trial	53	73 ± 7 *	Cognitive disorders
Markowitz, 2020 [16]	Canada	Controlled trial	30	76 ± 8 *	Macular degeneration
Chao, 2019 [17]	USA	Controlled trial	8	80 ± 6 *	Cognitive disorders
Maksimovich, 2019 [18]	Russia	Cohort	27	78 [58–81] ^†^	Binswanger’s disease
			62	77 [52–80] ^†^	Vascular parkinsonism
Brzak, 2018 [19]	Croatia	Controlled trial	30	72 [52–85] ^‡^	Hyposalivation
Merry, 2017 [20]	Canada	Cohort	24	78 ± 8 *	Macular degeneration
Saltmarche, 2008 [21]	Canada	Cohort	16	85 [76–97] ^†^	Chronic wounds
Lucas, 2003 [22]	Netherlands	Controlled trial	86	83 ± 9 *	Decubitus ulcer

Age: * mean ± standard deviation; ^†^ mean [range]; ^‡^ median [range].

**Table 2 biomedicines-12-01409-t002:** Characteristics of the photobiomodulation treatments.

Author, Year Ref	Device	Wave Length (nm)	Power Output (mW)	Irradiation Duration	Dose per Treatment (Joule/cm^2^)	Beam Spot Size	Pulse Frequency (Hz)	Target Location	Number of Sessions	Interval between Sessions
Neurodegenerative diseases										
Herkes, 2023 [13]	tPBM helmet ”Neuro”	635 (red) 810 (IR)	27 (red) 52 (IR)	12 min (red) 12 min (IR)	5.4 (red) 10.4 (NIR))	ND	ND	Transcranial	6 days per week for 12 weeks	1 day
Blivet, 2022 [15]	REGEnLIFE RGn530	660 (red) 850 (IR) 850 (laser)	25.5 (red) 28.8 (IR) 21.4 (laser)	25 min	19.1 (red) 21.57 (IR) 16.02 (laser)	929.6 mm^2^ (red) 94.8 mm^2^ (IR) 102.6 mm^2^ (laser)	10	Transcranial Abdomen	5 times per week for 8 weeks	2 days
Chao, 2019 [17]	Vielight Neuro Gamma Device	810	100 (posterior transcranial) 75 (anterior transcranial) 25 (intranasal LED)	20 min	120 (posterior) 90 (anterior) 30 (intranasal)	100 mm^2^	40	Transcranial Intranasal	3 times per week for 12 weeks	2–3 days
Maksimovich, 2019 [18]	Helium-neon laser ULF-01 Anod Ltd.	633	25–45	20–40 min	29–106	0.8–1.6 mm^2^	continuous	Intracerebral (intra-arterial)	1	NA
Wounds and ulcers										
Degerman, 2022 [14]	MID-LASER (Irradia)	635 (red) 904 (IR)	75 (red) 60 (IR)	30 + 120	2.4 (ulcer) 0.6 (intact skin)	9 mm^2^ (635 nm) 5.5 mm^2^ (904 nm)	250 (red) 700 (IR)	Venous Leg Ulcer (VLU) and intact skin close to VLU	2 times per week	2 and 3 days
Saltmarche, 2008 [21]	MedX Low Level Laser 1000 Console System	785	5–50	60 s (wound margin) 30 s (wound bed) 120 s (eschar)	1 to 6	ND	ND	Pressure ulcers Diabetes ulcers Venous ulcer At risk areas	Week 1: daily × 5 days Weeks 2–9: 3× weekly	Week 1: 1 day Weeks 2–9: 2–3 days
Lucas, 2003 [22]	ND	904	12 × 8	125 s	1	30 cm^2^	830	Wounds in different locations	5 days per week for 6 weeks	2 days
Macular degeneration										
Markowitz, 2020 [16]	LumiThera Valeda Light Delivery System	590 (yellow) 660 (red) 850 (NIR)	5 (yellow) 65 (red) 8 (NIR)	250 s	20.8 (yellow) 54.16 (red) 6.7 (NIR)	30 mm^2^	Pulse (yellow) Pulse (NIR) Continuous (red)	Eye	18	5 months
Merry, 2017 [20]	Warp 10 (Quantum devices) and Gentlewaves (Light Bioscience)	590 (yellow) 670 (red) 790 (NIR)	50–80 (red) 4 (yellow) 0.6 (NIR)	88 ± 8 s (red) 35 s (yellow) 35 s (NIR)	0.1 4–7.68 0.1	ND	2.5	Eye	9 sessions over 3 weeks	2–3 days
Hyposalivation										
Brzak, 2018 [19]	BTL-2000 Medical Technologies	685 (red) 830 (NIR)	30 (red) 35 (NIR)	5 min (red) 4:17 min (NIR)	1.8	ND	5.2	Parotide	10 consecutive days	1 day
				2 min (red) 1:43 min (NIR)	1.8	ND		Sub-mandibular		
				1 min (red) 51 s (NIR)	1.8	ND		Sublingual		

ND: not determined; NA: not appropriate; IR: infrared; NIR: near infrared.

**Table 3 biomedicines-12-01409-t003:** Description of the outcomes and results of photobiomodulation treatment in the 10 studies included in the systematic review.

Author, Year Ref	Study Design	Medical Condition	Primary Outcome	Group	Results	*p*
Neurodegenerative diseases						
Herkes, 2023 [13]	Controlled trial	Parkinson’s disease	MDS-UPDRS-III mean difference after stage 1	Sham	Mean: 3.9 95% CI: −3.5–11.3	NS
				Active		
			MDS-UPDRS-III Mean difference after stage 2	Sham	Mean: −3.1 95% CI: −10.6–2.7	NS
				Active		
Blivet, 2022 [15]	Controlled trial	Cognitive disorders	ADAS-Cog (absolute change)	No PBM	1.9 ± 4.1	0.49
				PBM	0.9 ± 4.9	
Chao, 2019 [17]	Controlled trial	Cognitive disorders	ADAS-Cog (week 12)	Usual care	39.2 ± 2.6	0.007
				PBM	32.3 ± 4.8	
			NPI-FS (week 12)	Usual care	20.3 ± 3.5	0.03
				PBM	13.5 ± 2.0	
Maksimovich, 2019 [18]	Cohort	Binswanger’s disease	% good clinical result (12–24 months)	Intervention	9/14	<0.005
				Control	0/13	
		Vascular parkinsonism	% good clinical result (12–24 months)	Intervention	9/37	<0.005
				Control	0/25	
Wounds and ulcers						
Degerman, 2022 [14]	Cohort	Venous leg ulcer	Healing time (days)	Intervention	68.8 ± 64.1	0.0002
				Control	431.5 ± 498.1	
Saltmarche, 2008 [21]	Cohort	Chronic wounds	Difference in Push Score (paired)	Pre/Post	−3.2 ± 3.4	0.003
Lucas, 2003 [22]	Controlled trial	Decubitus ulcer	Absolute wound size reduction (mm^2^)	Control	138 ± 270	0.23
				PBM	48 ± 394	
			Relative wound size reduction (%)	Control	34 ± 204	0.42
				PBM	5 ± 194	
Macular degeneration						
Markowitz, 2020 [16]	Controlled trial	Macular degeneration	BCVA letter score at month 1	Sham	1.2 ± 5.4	0.10
				PBM	3.8 ± 5.1	
			BCVA letter score at month 7	Sham	1.7 ± 6.0	0.16
				PBM	4.3 ± 6.2	
Merry, 2017 [20]	Cohort	Macular degeneration	BCVA letter score	BL to V1	+5.90	<0.001
				BL to V2	+5.14	<0.001
Hyposalivation						
Brzak, 2018 [19]	Controlled trial	Hyposalivation	Salivary flow rate	830 nm	0.20 mL/min (Day 1) to 0.35 mL/min (Day 10)	0.0019
				685 nm	0.15 mL/min (Day 1) to 0.25 mL/min (Day 10)	0.0044

MDS-UPDRS-III, Movement Disorders Society revision of the Unified Parkinson’s Disease Rating Scale Part III motor scale; ADAS-COG, Alzheimer’s Disease Assessment Scale–Cognitive Subscale; NPI-FS: Neuropsychiatric Inventory (Frequency × Severity); BCVA: Best-corrected Visual Acuity; PBM: photobiomodulation; BL: Baseline; V1; Visit 1; V2: Visit 2.

**Table 4 biomedicines-12-01409-t004:** (a): Quality assessment of the studies included in the systematic review using the Newcastle Ottawa Scale (NOS)—observational studies. (b): Quality assessment of the studies included in the systematic review using the Cochrane RoB tool—randomized controlled studies.

Author, Year Ref	Selection	Comparability	Outcome	Total Score	Quality Rating	
Degerman, 2022 [14]	****	-	***	7	High	
Maksimovich, 2019 [18]	****	-	***	7	High	
Merry, 2017 [20]	****	-	***	7	High	
Saltmarche, 2008 [21]	****	**	***	9	High	
	**Risk of Bias**					
**Author, Year Ref**	**Randomization Process**	**Deviation from Intended Intervention**	**Missing Outcome Data**	**Measurement of the Outcome**	**Selection of the Reported Result**	**Overall Bias**
Herkes, 2023 [13]	Low	Low	Low	Low	Low	Low
Blivet, 2022 [15]	Low	Low	Low	Low	Low	Low
Markowitz, 2020 [16]	Low	Low	Low	High	High	High
Chao, 2019 [17]	Low	Some concerns	Low	High	High	High
Brzak, 2018 [19]	Low	Some concerns	Low	Low	Low	Low
Lucas, 2003 [22]	Low	Low	Low	Low	Low	Low

NOS scores of ≥7 were considered as high quality, 5–6 as moderate quality, and NOS scores < 5 as low quality. RoB: Risk of Bias (Cochrane library tool). *: each star counts for one point (the total score is the sum of the number of stars in a row)

## Data Availability

The data can be made available upon reasonable request at moustapha.drame@chu-martinique.fr.

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
