# Peer review of "Efficacy of Photobiomodulation Therapy in Older Adults: A Systematic Review"

_biomedicines, 2024, doi:10.3390/biomedicines12071409_

Round 1

Reviewer 1 Report

Comments and Suggestions for Authors

This is a pertinent and well performed research. Discussion is quite good and highlight strengths and limitations.

Nevertheless, some comments:

Mention to PRISMA methodology needs a reference.

Please, add a PICO question.

Author Response

Authors: We thank the reviewer for the expertise and valuable advice, which have helped to improve our manuscript

R1# Mention to PRISMA methodology needs a reference.

Authors: A reference has been added for the PRISMA guidelines.

R1# Please, add a PICO question.

Authors: The appropriate PICOS question (Population, Intervention, Comparator, Outcome, and Study design) has been added to the Methods section.

Reviewer 2 Report

Comments and Suggestions for Authors

The manuscript by Moustapha Drame et al. relates to the evaluation of the efficacy of photobiomodulation therapy in the elderly. The logical structure of the manuscript is adequate. The language is understandable, but some errors should be corrected.  Although the manuscript is within the scope of Biomedicine Journals, I am not sure if it is within the scope of the Molecular and Translational Medicine section to which it was submitted. This section is devoted to studies of disease pathogenesis at the molecular or physiological level, as well as basic preclinical, clinical & translational aspects.

Major comments:

1) According to the authors, can the preparation of a review based on 10 references be considered a comprehensive literature review? I am not quite sure that the analysis of only 10 references can be considered a comprehensive review, even a systematic one, especially considering the Biomedicines Journal Board recommendation that "Reviews provide a comprehensive analysis of the existing literature within a field of study, identifying current gaps or problems. Simply searching and then rejecting most of the search results and preparing a review based on 10 publications, which represent less than 2.5% of all the literature initially searched, is in my opinion definitely not sufficient to publish such a review in a journal with an impact factor of 4.7.  

2) "A comprehensive literature search was performed in Scopus© and PubMed©" - why did the authors decide to use only these two platforms for the search?  

3) What was the criterion for excluding the 285 references initially found? What was the criterion for excluding them based on titles and abstracts read?

4) In the context of the content of the manuscript, the choice of patient age group is, in my opinion, unjustified. Why did the authors decide to analyse the effectiveness of photobiomodulation therapy (PBMT) in older adults (65 years and older)? In general, the PBMT is more extensively used in the treatment of younger patients. On the beginning I supposed that the choice of the age group is related to the possible effect on the symptoms of Alzheimer’s disease, as it was stated in the introduction section: ” PBMT is also being discussed as a possible approach for disease modification in Alzheimer’s disease by the EU/US CTAD Task Force [9]. In light of these developments, the role of photobiomodulation in the care of individuals aged 65 years and over is garnering increasing interest.” However, after selection of the publication for review, only 4 out of 10 analysed publications are related to the neurodegenerative diseases. If, the authors found only 4 publications related to the application of the PBMT in neurodegenerative diseases in this range of time, the choice of the age group is in my opinion questionable.   The vast majority of remained publications used in this examination deal with other diseases, not necessarily related to this age group. Therefore, I don’t understand the choice of so limited age group for purpose of this review. Particularly, if the choice of this group is significantly limiting the number of possible publications related to the PBMT application.

5) Is the sample size of 8 patients in Chao, 2019 [16] sufficient and associated with statistically significant results compared to other selected publications in Table 1?

6) "In all the studies included in this systematic review, the protocols used to achieve photobiomodulation were different in terms of type of device, wavelength, irradiation duration, pulse frequency, etc" - In PBMT or LLLT, the most important therapeutic parameter is the delivered irradiation energy dose, energy density or fluence [J/m2; W/m2], which influences the effects induced in the tissue: photothermal, photochemical, photomechanical or photobiological. The authors compare the various continuous and pulsed radiation sources of different wavelengths (see Table 2), but they have not even bothered to analyse, study and discuss this most important parameter in PBMT. In my opinion, without this parameter, any comparison or analysis of the results is meaningless and of no value to the reader.

Minor comments:

) There are quite a few vague statements in English in the manuscript  e.g. “ specific length of time” – this expression is confusing. Time and length are different physical quantities, so don't mix it up like that. It should be better " for specific period of time".

2) It would have been valuable if the authors had introduced line numbering in the manuscript, which would have made the reviewer's work and reference to sections of the manuscript more efficient.

Author Response

The manuscript by Moustapha Drame et al. relates to the evaluation of the efficacy of photobiomodulation therapy in the elderly. The logical structure of the manuscript is adequate. The language is understandable, but some errors should be corrected. Although the manuscript is within the scope of Biomedicine Journals, I am not sure if it is within the scope of the Molecular and Translational Medicine section to which it was submitted. This section is devoted to studies of disease pathogenesis at the molecular or physiological level, as well as basic preclinical, clinical & translational aspects.

Authors: The manuscript has been proofread by a native English speaking medical writer. We are confident that the quality of the English is now beyond reproach, but we remain open to any specific suggestions for correction.

Concerning the section, we defer to the journal's choice regarding the inclusion of our article in the Molecular and Translational Medicine section.

Major comments:

1) According to the authors, can the preparation of a review based on 10 references be considered a comprehensive literature review? I am not quite sure that the analysis of only 10 references can be considered a comprehensive review, even a systematic one, especially considering the Biomedicines Journal Board recommendation that "Reviews provide a comprehensive analysis of the existing literature within a field of study, identifying current gaps or problems. Simply searching and then rejecting most of the search results and preparing a review based on 10 publications, which represent less than 2.5% of all the literature initially searched, is in my opinion definitely not sufficient to publish such a review in a journal with an impact factor of 4.7. 

Authors: Regarding the number of studies included in the systematic review (n=10), we fully understand the Reviewer’s concerns and we agree, in part, with the remark. However, despite the existence and numerous and varied articles about photobiomodulation and their positive effects in different pathologies, an international panel of clinicians and researchers could only issue a few recommendations, due to the diversity and the level of evidence of the studied articles (Robijns et al., 2022). What may at first glance appear to be a limitation (including only 10 articles) may actually be of particular interest in this context, especially in the older population. In situations of therapeutic innovation, older people are often the last to benefit, and are often excluded from trials testing new therapeutic approaches. Therefore, a systematic review comprising 10 articles is already an important source of information in this field. There are numerous other systematic reviews in the literature that included a similar number of articles, illustrating the complexity of thematic reviews on this topic (ref Petz et al, 2020, journal IF 3.3, Bensadoun et al 2020, journal IF 4).

2) "A comprehensive literature search was performed in Scopus© and PubMed©" - why did the authors decide to use only these two platforms for the search? 

Authors: We used the Pubmed and Scopus platforms for the literature search because, firstly, they cover the vast majority of the medical literature, and secondly, they are the two bibliographic databases to which we have unlimited access. However, as this may be seen as a limitation, we have now mentioned it in the "Limitations" section of the discussion.

3) What was the criterion for excluding the 285 references initially found? What was the criterion for excluding them based on titles and abstracts read?

Authors: The 285 studies excluded on the basis of title and/or abstract were excluded because the population was clearly inappropriate (young subjects) or the type of study did not comply with the selection criteria (Correspondence, editorials, reviews, basic science articles, and case reports, as well as case series of less than 10 subjects).

4) In the context of the content of the manuscript, the choice of patient age group is, in my opinion, unjustified. Why did the authors decide to analyse the effectiveness of photobiomodulation therapy (PBMT) in older adults (65 years and older)? In general, the PBMT is more extensively used in the treatment of younger patients. On the beginning I supposed that the choice of the age group is related to the possible effect on the symptoms of Alzheimer’s disease, as it was stated in the introduction section:” PBMT is also being discussed as a possible approach for disease modification in Alzheimer’s disease by the EU/US CTAD Task Force [9]. In light of these developments, the role of photobiomodulation in the care of individuals aged 65 years and over is garnering increasing interest.” However, after selection of the publication for review, only 4 out of 10 analysed publications are related to the neurodegenerative diseases. If, the authors found only 4 publications related to the application of the PBMT in neurodegenerative diseases in this range of time, the choice of the age group is in my opinion questionable.   The vast majority of remained publications used in this examination deal with other diseases, not necessarily related to this age group. Therefore, I don’t understand the choice of so limited age group for purpose of this review. Particularly, if the choice of this group is significantly limiting the number of possible publications related to the PBMT application.

Authors: The choice of age group is based on the fact that we are geriatricians/gerontologists and our aim was to review the state of the art in PBMT, which is still underused in older populations. The value of PBMT in older persons needs to be explored in all its aspects, not just in neurodegenerative diseases. Some authors (Florisson S.  Basic Clin Pharmacol Toxicol, 2021; Vieujean S. Lancet Healthy Longevit. 2022) have previously underlined that older people are under-represented in clinical research. This fact can often lead to the use of inappropriate or ineffective treatments in the older population (Seegers Plos 2013)), in view of their comorbidities or physiological specificities. We aim to contribute to the wider development of PBMT in older populations for many different indications, since several studies attest to the safety of PBM and the absence of contra-indications, making it an attractive therapeutic option. Adapting protocols to the specificities of older adults would probably be the first step to success. Clearly, the small number of publications relating to PBMT use specifically in older people is a reflection of the knowledge gap that persists. Therefore, our review of the state of the art shows that there is a compelling need for more studies in this area.

5) Is the sample size of 8 patients in Chao, 2019 [16] sufficient and associated with statistically significant results compared to other selected publications in Table 1?

Authors: It is true that the study by Chao included only 8 subjects. However, this study met all the selection criteria for inclusion in the systematic review. In addition, despite the small sample size, this study showed significant differences (as shown in Table 3). In addition, we did not perform any meta-analysis of individual results, so the small number of subjects included in Chao’s study is not a limitation for its inclusion in our review.  

 6) "In all the studies included in this systematic review, the protocols used to achieve photobiomodulation were different in terms of type of device, wavelength, irradiation duration, pulse frequency, etc" - In PBMT or LLLT, the most important therapeutic parameter is the delivered irradiation energy dose, energy density or fluence [J/m2; W/m2], which influences the effects induced in the tissue: photothermal, photochemical, photomechanical or photobiological. The authors compare the various continuous and pulsed radiation sources of different wavelengths (see Table 2), but they have not even bothered to analyse, study and discuss this most important parameter in PBMT. In my opinion, without this parameter, any comparison or analysis of the results is meaningless and of no value to the reader.

Authors: We thank the Reviewer for raising this very important point. We have now listed the energy dose in Table 2. We have added a discussion of this parameter for each medical condition in the Discussion section. Of note, this point was already discussed in the paragraph on wounds and ulcers.

In each study, the outcome of the group with PBMT was compared with the outcome of the group without PBMT, and unfortunately, not according to the energy dose delivered.  We agree with the Reviewer that it would have been interesting to compare protocols head-to-head. Indeed, harmonization across studies is necessary, as underlined by Hamblin et al in their article entitled “How to write a good article in photobiomodulation”.

Minor comments:

1) There are quite a few vague statements in English in the manuscript e.g. “specific length of time” – this expression is confusing. Time and length are different physical quantities, so don't mix it up like that. It should be better " for specific period of time".

Authors: Thank you for pointing this out. The paper has been thoroughly reviewed by a native English-speaking medical writer. Appropriate modifications have been made accordingly.

2) It would have been valuable if the authors had introduced line numbering in the manuscript, which would have made the reviewer's work and reference to sections of the manuscript more efficient.

Authors: Thank you for this suggestion. We fully understand the Reviewer's discomfort, and agree that line numbering is useful. However, we respected the journal's recommendations to authors with regard to the format of the manuscript, and line numbering is not recommended.

Reviewer 3 Report

Comments and Suggestions for Authors

The manuscript is well written and the outlines are very clear
The aim of this review was to determine whether there is any available evidence on the efficacy of photobiomodulation therapy (PBMT) in older adults. About the studies 10 records were included in the final review. In all included studies, the protocols used to deliver PBMT were different in terms of type of device, wavelength, irradiation duration, and pulse frequency. In neurodegenerative diseases, two studies reported non-significant results, while two studies reported efficacy of PBMT. In wounds and ulcers, two out of three studies reported efficacy of PBMT. In macular degeneration, one study reported efficacy of PBMT. The only one study on hyposalivation reported efficacy of PBMT. The authors concluse that PBMT appears to be a promising complementary treatment. but in the future it will be essential to harmonize PBMT parameters therefore further studies are warranted to define the best indications, the most effective protocols, and the right population to target, for use in routine practice.

To capture the reader's attention, the manuscript must delve deeply into the mechanisms of PBMT. Nothing is mentioned about the cellular and molecular mechanisms evoked by this effect. It is not enough to just mention the physical mechanisms but this topic is of broad importance and many obscure points must be clarified. Furthermore, it is very interesting that the authors consider the efficacy of PBMT in elderly patients and therefore (again) the cellular and molecular mechanisms evoked must necessarily be clarified. Furthermore, the light sources considered are not clear, lasers, LEDs, photodynamic therapy to give just a few examples. It follows that the manuscript must necessarily be revised based on these suggestions.

Comments on the Quality of English Language

Minor editing of English language required

Author Response

The manuscript is well written and the outlines are very clear

Authors: Thank you for your positive appraisal and useful suggestions for improvement.

The aim of this review was to determine whether there is any available evidence on the efficacy of photobiomodulation therapy (PBMT) in older adults. About the studies 10 records were included in the final review. In all included studies, the protocols used to deliver PBMT were different in terms of type of device, wavelength, irradiation duration, and pulse frequency. In neurodegenerative diseases, two studies reported non-significant results, while two studies reported efficacy of PBMT. In wounds and ulcers, two out of three studies reported efficacy of PBMT. In macular degeneration, one study reported efficacy of PBMT. The only one study on hyposalivation reported efficacy of PBMT. The authors concluse that PBMT appears to be a promising complementary treatment. but in the future it will be essential to harmonize PBMT parameters therefore further studies are warranted to define the best indications, the most effective protocols, and the right population to target, for use in routine practice.

To capture the reader's attention, the manuscript must delve deeply into the mechanisms of PBMT. Nothing is mentioned about the cellular and molecular mechanisms evoked by this effect. It is not enough to just mention the physical mechanisms but this topic is of broad importance and many obscure points must be clarified. Furthermore, it is very interesting that the authors consider the efficacy of PBMT in elderly patients and therefore (again) the cellular and molecular mechanisms evoked must necessarily be clarified. Furthermore, the light sources considered are not clear, lasers, LEDs, photodynamic therapy to give just a few examples. It follows that the manuscript must necessarily be revised based on these suggestions.

Authors: Thank you for raising these pertinent points. We have taken these suggestions into account in the revised manuscript.

Round 2

Reviewer 2 Report

Comments and Suggestions for Authors

I thank the authors for their clarifications and corrections made to the manuscript.

Although the reviews cited by the authors (ref. Petz et al, 2020, journal IF 3.3, Bensadoun et al 2020, journal IF 4) were published on the basis of a similar number of publications analysed, they were much more extensive, with additional considerations: the use of laser therapy in a much larger number of diseases, or considerations of the methodology of the study (choice of radiation sources, doses, instrument for evaluating the effectiveness of the therapy).

On the other hand, the manuscript under review dealt with these issues in a rather superficial manner.

Although I appreciate the authors' efforts, in reality the use of laser therapy, especially photobiomodulation therapy, is not widespread enough, which I regret. The authors could have enriched their work by considering, for example, the relationship between changes in the optical properties of tissues in the elderly and the efficacy of phototherapy.

Reviewer 3 Report

Comments and Suggestions for Authors

The authors have asked correctly to my questions

Comments on the Quality of English Language

Minor editing of English language required